# Inactivation of Dermatophytes Causing Onychomycosis and Its Therapy Using Non-Thermal Plasma

**DOI:** 10.3390/jof6040214

**Published:** 2020-10-10

**Authors:** Jaroslav Lux, Radim Dobiáš, Ivana Kuklová, Radek Litvik, Vladimír Scholtz, Hana Soušková, Josef Khun, Jakub Mrázek, Michaela Kantorová, Pavla Jaworská, Táňa Prejdová, Jana Šnupárková, Petr Hamal, Jaroslav Julák

**Affiliations:** 1Department of Epidemiology and Public Health, Faculty of Medicine, University of Ostrava, Syllabova 19, 703 00 Ostrava, Czech Republic; jaroslav.lux@seznam.cz; 2Podiatric Center Medicia, Daliborova 421/15, 709 00 Ostrava, Czech Republic; 3Department of Bacteriology and Mycology, Public Health Institute in Ostrava, Partyzánské nám. 7, 702 00 Ostrava, Czech Republic; radim.dobias@seznam.cz (R.D.); jakub.mrazek@zuova.cz (J.M.); michaela.kantorova@zuova.cz (M.K.); pavla.jaworska@zuova.cz (P.J.); tana.ryskova@zuova.cz (T.P.); 4Department of Biomedical Sciences, Institute of Microbiology and Immunology, Faculty of Medicine, University of Ostrava, Syllabova 19, 703 00 Ostrava, Czech Republic; 5Department of Dermatology and Venereology, First Faculty of Medicine, Charles University and General University Hospital in Prague, U Nemocnice 499/2, 128 08 Praha 2, Czech Republic; ivana.kuklova@lf1.cuni.cz; 6Department of Dermatology, University Hospital Ostrava, 17. listopadu 1790/5, 708 52 Ostrava, Czech Republic; radek.litvik@fno.cz; 7Department of Physics and Measurements, Faculty of Chemical Engineering, University of Chemistry and Technology Prague, Technická 5, 166 28 Praha, Czech Republic; josef.khun@gmail.com; 8Department of Computing and Control Engineering, Faculty of Chemical Engineering, University of Chemistry and Technology Prague, Technická 5, 166 28 Praha, Czech Republic; Hana.Souskova@vscht.cz; 9Department of Mathematics, University of Chemistry and Technology Prague, Technická 5, 166 28 Praha, Czech Republic; snuparkj@vscht.cz; 10Department of Microbiology, University Hospital Olomouc, Hněvotínská 3, 775 15 Olomouc, Czech Republic; petr.hamal@fnol.cz; 11Institute of Immunology and Microbiology, First Faculty of Medicine, Charles University, Studničkova 7, 128 00 Praha 2, Czech Republic; jaroslav.julak@lf1.cuni.cz

**Keywords:** non-thermal plasma, *Trichophyton*, fungal inactivation, onychomycosis therapy

## Abstract

Onychomycosis is one of the most common nail disorders. Its current treatment is not satisfactorily effective and often causes adverse side effects. This study aims to determine the optimal conditions for non-thermal plasma (NTP) inactivation of the most common dermatophytes in vitro and to apply it in patient`s therapy. The in vitro exposure to NTP produced by negative DC corona discharge caused full inactivation of *Trichophyton* spp. if applied during the early growth phases. This effect decreased to negligible inactivation with the exposure applied six days after inoculation. In a group of 40 patients with onychomycosis, NTP therapy was combined with nail plate abrasion and refreshment (NPAR) or treatment with antimycotics. The cohort included 17 patients treated with NPAR combined with NTP, 11 patients treated with antimycotics and NTP, and 12 patients treated with NPAR alone. The combination of NPAR and NTP resulted in clinical cure in more than 70% of patients. The synergistic effect of NPAR and NTP caused 85.7% improvement of mycological cure confirmed by negative microscopy and culture of the affected nail plate. We conclude that NTP can significantly improve the treatment of onychomycosis.

## 1. Introduction

Some fungi cause serious or fatal diseases or intoxication, with the genus *Aspergillus* being a typical example. Dermatophytes are less dangerous, but they are important causative agents of benign, but bothersome superficial mycoses in humans. This group includes mainly the genera *Trichophyton* and *Microsporum* containing approximately ten species of parasitic fungi. They may be transmitted from domestic and farm animals (zoophilic dermatophytes) or the environment (geophilic dermatophytes); sometimes, the category of anthropophilic dermatophytes is also recognized. Infections in animals are often asymptomatic, whereas infections in human manifest as inflammatory infections of the skin (tinea corporis, tinea faciei, tinea pedis), hairy parts of the head (tinea barbae, tinea capitis), or nails (tinea unguium or onychomycosis). The latter category is mainly caused by the anthropophilic species *Trichophyton rubrum*, *T. tonsurans*, and *Epidermophyton floccosum*, less frequently by zoophilic *T. interdigitale*, *T. benhamiae*, *Microsporum canis*, and *T. verrucosum.* Geophilic species such as *Nannizzia gypsea* and *N. persicolor* are only occasionally responsible.

As far as therapy is concerned, topically applied preparations based on imidazoles, allylamines, pyridines, or morpholines are suitable for tinea corporis, namely clotrimazole, miconazole, econazole, ketoconazole, oxiconazole, tioconazole, terbinafine, naftifine, ciclopirox olamine, and amorolfine. Treatment of tinea capitis is more difficult, and systemic oral administration of antifungals such as terbinafine is usually required. Systemic application of itraconazole, fluconazole, posaconazole, or griseofulvin is problematic due to various limitations.

For onychomycosis, topical application of the above antifungal nail polish and solutions may be sufficient, but oral administration is sometimes necessary, although it is complicated by side effects. A nail debridement involving removal of the affected parts of the nail can be used as auxiliary therapy. In general, however, the therapy is a long-lasting process (one year or more), and the efficacy of treatment is low; only 50% successfully cured cases are reported. Elewski (1998) [1], Roberts et al. (2003) [2], Gupta et al. (2017) [3], and Asz-Sigall et al. (2017) [4] described this issue in detail.

The relatively low efficiency of the classical therapy has motivated efforts to apply new physical therapy methods. Among them, the experiments with heating the nail to approximately 40–50°C with a Nd:YAG laser was described and its use carefully reviewed by Bristow (2014) [5] and Francuzik et al. (2016) [6]. However, this method has not been proven to be suitable for patient therapy. An attempt to cure onychomycosis by photodynamic therapy using illumination with LED at 635 nm and 37 J cm^−2^ was described by Gilaberte et al. (2011) [7]. In combination with nanoemulsions, photodynamic therapy proved to be effective in 60% of 20 cases [8]. The practical efficacy of other methods, especially iontophoresis and ultrasound, has been rather skeptically evaluated in the literature [9].

Recently, low-temperature plasma (non-thermal plasma (NTP)) applications for mold inactivation have been reported. Plasma, also called the fourth state of matter, is a partially or fully ionized gas. There is a distinction between high-temperature plasma, reaching temperatures of thousands of Kelvin, and NTP, which occurs at nearly ambient temperature and contains low-temperature ions and highly energetic free electrons. NTP is a partially ionized gas where most of the energy is stored in the kinetic energy of the electrons, whereas ions remain at room temperature. This ionized gas represents a cold mixture of free radicals and charged particles and does not increase the temperature of the material on which it is applied. NTP may be easily obtained by various electric discharge burning at or between point or plane electrodes with high voltage potential where free electrons are accelerated by an electric field and generate the secondary electrons, ions, and photons by collisions with neutral particles. The most commonly used discharges are corona discharges, plasma jet (also called plasma needle, plasma torch, or plasma pen), dielectric barrier discharge, gliding arc, and microwave discharges. A special DC discharge called cometary was described by Scholtz and Julák (2010, 2010a) [10,11]. For a more detailed description of plasma sources, see for example Yousfi et al. (2011) [12], Khun et al. (2018) [13], or Julák et al. (2018) [14].

The microbicidal activity of NTP is mediated mainly by reactive oxygen particles and reactive nitrogen particles arising from the surrounding gases. Various species such as ions, radicals, and stable or unstable electroneutral molecules, namely superoxide anion, singlet oxygen, hydroxyl and hydroperoxyl radical, nitric oxide radical, peroxynitrite, and others, are present. The lifetimes of these species are very short, with typical half-lives ranging from nanoseconds to a few seconds. The stable compounds formed are hydrogen peroxide, ozone, and nitrogen oxides. For details, see Graves (2012) [15], Kelly and Turner (2013) [16], Sysolyatina et al. (2014) [17], or Liu et al. (2016) [18]. The mechanisms of the biological effects of NTP in unicellular microbes are still poorly understood; apart from physical destruction and necrosis, apoptosis also occurs in unicellular microbes including yeasts. As described by Lunov et al. (2016) [19], the exposure of bacteria or yeasts to NTP not only induces direct physical destruction, but also triggers programmed cell death and apoptosis. Some hallmarks of apoptosis were also found in higher unicellular eukaryotes such as *Trypanosoma* spp. or *Dictyostelium discoideum*.

NTP is widely used in many areas of human activity including the modification of the surface of various materials (surface termination, increasing of wettability), the food industry (food decontamination, increase of seed wettability), biotechnology (microbial decontamination), wastewater treatment, biology, and medicine (wound and skin infection healing, blood coagulation); for a more detailed description of its applications, reviews by Tendero et al. (2006) [20], Julák and Scholtz (2020) [21], Zhao et al. (2020) [22], or the comprehensive book by Metelmann et al. (2018) [23] may be recommended. Medical applications mainly include disinfection processes, but also acceleration of blood coagulation and improved wound healing, dental applications, or cancer therapy [24,25].

Most studies on the disinfection effects of NTP were devoted to bacteria, but attempts to inactivate fungi both in vitro and in vivo have also been reported [26,27,28,29,30,31]. Thus, this issue is sufficiently and extensively evaluated, and the results show the possibility of effective inactivation of fungi by NTP. However, the range of experimental parameters and specific results is rather wide. For example, Misra et al. (2019) [31] reported exposure times required to inactivate *Aspergillus* spp. of 5, 9, 10, and 15 min, as well as only 15 seconds in one case.

The results of our earlier efforts to clarify this issue served as the basis for the present communication. In general, different microbes exhibited different sensitivity to NTP; while bacteria could be completely inactivated within seconds to minutes, yeasts required exposure for several minutes and mold spores for tens of minutes. Comparable exposure times require microorganisms in the form of a biofilm, as these are considerably more resistant to the microbicidal action of plasma in comparison with their planktonic forms [32]. Significant differences were also observed between mold species. For example, *Cladosporium sphaerospermum* spores were completely inactivated within 10 min, whereas *Aspergillus oryzae* spores were not inactivated even after 40 min under the same conditions; *Alternaria* spp. and *Byssochlamys nivea* exhibited intermediate sensitivity [14]. Soušková et al. (2011) [33] presented similar results. Whereas total inactivation of yeast occurred in six minutes, spores needed 20–25 min of exposure in the case of *Cladosporium sphaerospermum* and *Penicillium crustosum*; *Aspergillus oryzae* spores were not completely inactivated even after 30 min of exposure. Scholtz et al. (2015) [34] described the sensitivity of dermatophytes. The anthropophilic and zoophilic species *Trichophyton rubrum* and *T. interdigitale* were found to be highly sensitive to NTP both in suspension and on surfaces, and so was zoophilic *Arthroderma benhamiae*. In contrast, the geophilic species *Nannizzia gypsea* appeared as highly resistant. In all these studies, significant differences were observed between various modes of NTP production, namely between positive and negative DC corona or between corona and dielectric barrier discharge [14].

Xiong et al. (2016) [35] suggested the possible therapy of onychomycosis by NTP in a model of bovine hoof slices infected by *Trichophyton rubrum*. Daeschlein et al. (2010) [36] found NTP as a supportive and/or alternative antimycotic tool in tinea pedis treatment. Therapy of superficial fungal skin infection using NTP was also verified on guinea pigs artificially infected with *Trichophyton mentagrophytes* [37]. NTP also appeared useful in the treatment of human tinea corporis caused by *Trichophyton interdigitale* [38].

Several clinical trials of NTP therapy of human onychomycosis are currently underway [39,40,41]. Different approaches to NTP application are methodologically interesting. The exposure times reported in the studies were as follows: 20 min once a week for three weeks; three doses per week for 14 days, then once a month (exposure time not specified); three exposures lasting 45 min in one week. Preliminary results of our clinical study were published by Lux et al. (2018) [42]. Patients were treated with NTP in 16 exposures of 20 min each, after which the dermatophyte was no longer present, as confirmed by microscopy, detection of DNA, and culture. The preliminary data show that the exposure of the affected nail to NTP alone is not sufficient and must be combined with nail plate abrasion and refreshment (NPAR).

For NTP applications in human medicine, it is important that it does not cause any adverse changes to the skin. This fact was confirmed by numerous studies such as those by Julák and Scholtz (2013) [43], Haertel et al. (2014) [44], and Heinlin et al. (2010) [45].

In this communication, we present data on the dynamics of dermatophyte inactivation, which may explain the mechanism of NTP action and determine the optimal conditions of NTP exposure. The second part of the paper is devoted to the results of the study of human onychomycosis therapy using NTP.

## 2. Materials and Methods

### 2.1. In Vitro Experiments

#### 2.1.1. Fungal Strains

The following fungi were employed: *Trichophyton rubrum*, *Trichophyton interdigitale*, and *Trichophyton benhamiae* obtained from clinical material of the Public Health Institute in Ostrava. The dermatophyte strains were identified as etiological agents of onychomycosis and isolated from toenails. The experimental fungal strains used in the study were chosen according to the current epidemiological situation in the Czech Republic where *Trichophyton rubrum* and *T. interdigitale* are the most common anthropophilic and zoophilic agents of onychomycosis, respectively [46,47]. Isolates of *Trichophyton benhamiae* were added to this experiment as a marginal zoophilic etiological agent of onychomycosis.

#### 2.1.2. Cultures

All strains were pre-cultured on Sabouraud agar at 26 °C for 2 days. Suspensions of the fungi were prepared in sterile water with Tween 80 and vortexed. Clear supernatants were adjusted to a final concentration of fungal spores of 200 colony forming units per 100 μL and used as stock suspensions. Aliquots of 100 µL of these stock suspensions were then inoculated onto the entire surface of the experimental Sabouraud agar plates, which were subsequently exposed to NTP as described below.

#### 2.1.3. Arrangement of Experimental Plates

Plates inoculated with *T. rubrum*, *T. interdigitale*, and *T. benhamiae* were exposed to NTP according to the scheme depicted in Figure 1. The plates were marked with serial numbers indicating the order of inoculation and exposure. Number 1 denotes the plate inoculated and exposed on the first day of the experiment; Number 12 denotes the plate inoculated and exposed on the first and subsequently further exposed on the second day of the experiment, etc. Similarly, Numbers 3, 4, 5, 6, and 7 indicate the plate inoculated on Day 1 and first exposed on Day 3, with further exposures of growing cultures taking place on Days 4, 5, 6, and 7. Symbol ‘-’ indicates days on which no further exposure took place; the label 2---- thus identifies the plate inoculated on Day 1 and exposed on Day 2 only. All cultures were incubated at 26 °C throughout the experiment.

#### 2.1.4. NTP Exposure of Plates

NTP was produced by a negative DC corona discharge, burning in the apparatus schematically depicted in Figure 2. The working electrode was made from a medical injection needle; the other annular electrode of 12 mm in diameter was made of brass. The electrodes were connected to the voltage source HT 2103 (Utes Brno), held at 7 kV and 150 µA. An embedded metallic grid improved the discharge effect (see Scholtz et al. 2013 [48]). Based on our recent experimental results [37], the exposure time was two minutes in all cases.

#### 2.1.5. Evaluation

Growth cultures were evaluated in two ways. Colonies of *Trichophyton* spp. were counted visually every day and their numbers (colony forming units) were compared with unexposed cultures.

### 2.2. Onychomycosis Clinical Study

A multicenter study of therapeutic interventions for onychomycosis included a cohort of 40 randomly selected patients treated in centers listed in the annotation of this article. The patients with toenail onychomycosis were included based on moderate to severe onycholysis clinically confirmed with a dermatoscope and/or dermatophytic hyphae detected under a microscope and/or dermatophyte-positive cultures. The study adhered to the Declaration of Helsinki, 2013, and good clinical practice and was approved by the Ethics Committee for Multicenter Clinical Trials of the University Hospital Olomouc (NU20-05-00176, approval date 17. 6. 2019).

#### 2.2.1. Clinical Sampling and Mycological Examination

Onycholytic nails were cleaned with 70% isopropyl alcohol to prevent colonizing organisms from confounding culture results and to inhibit the growth of the relevant pathogen. The nail plate was then clipped back, and the subungual debris (preferred) was scraped into a sterile tube. The pieces of scraped onycholytic nails were transferred with a small curette onto a glass slide for microscopic examination. To dissolve larger keratinocyte material and determine the presence or absence of fungal elements, ten percent potassium hydroxide (KOH) was used with Myko-Ink stain (see Figure 3e,f).

The causative fungus and its viability were definitively identified through fungal culture, one of the criteria standards in diagnosing onychomycosis. Nail samples were cultured on two types of media: agar containing cycloheximide to inhibit non-dermatophytes and encourage growth of dermatophytes and agar without cycloheximide to culture non-dermatophytes. The cultures grew at 26 °C for up to a month. *T. rubrum* and *T. interdigitale* were identified according to macro- and micro-morphology characteristics (see Figure 3a–d).

#### 2.2.2. PCR

The long time needed to identify the dermatophyte and the high false negative rate are the limitations of culture techniques. For this reason, we added the PCR method combined with high-resolution melting analysis (HRMA) for dermatophyte detection and identification directly from clinical samples of the skin and its adnexa [49,50].

The ZR Fungal/Bacterial DNA MiniPrep kit (Zymo Research, Irvine, CA, USA) was used for fungal rDNA isolation from clinical material. The isolation was carried out according to the manufacturer’s instructions. The first step was to disrupt specimens of the clinical material using zirconium beads. DNA isolation was followed by buffers and isolation columns, and the DNA obtained was used for real-time PCR. Two PCR methods were used to detect rDNA using primer pairs capturing a broad spectrum of dermatophytes. The first method was targeted to amplify the ITS1-5,8S region, while the second method amplified the complete ITS rDNA region (ITS1-5,8S-ITS2). The sequences of the primers and probes used are shown in Table 1. Both analyses were performed on the LightCycler 96 instrument (Roche, Basel, Switzerland). Amplification of the ITS1-5,8S segment was performed under the following conditions: The PCR reaction was performed at a final volume of 20 µL. The mixture contained 5 µL of DNA from the clinical specimen and 15 µL of reaction mix containing 10 µL of TaqMan Fast Advanced Master Mix (Thermo Fisher Scientific, Waltham, MA, USA), 10 pmol of each pair of primers, and 4 pmol of the TaqMan probe. The amplification temperature profile included uracil-DNA glycosylase activation (50 °C/2 min), initial denaturation of 95 °C/20 s, and 40 amplification cycles consisting of 95 °C/3 s denaturation, annealing including elongation, and signal measurement at 60 °C. Evaluation of the reaction was qualitative; positive isolates were further forwarded to species identification by the real-time PCR-HRMA method. The real-time PCR-HRMA region of ITS1-5,8S-ITS2 was used for identification of the most important species. Amplification of the ITS1-5,8S-ITS2 segment was performed under the following conditions: PCR reaction was performed at a final volume of 20 µL; the mixture contained 5 µL of DNA from the clinical sample and 15 µL of reaction mix containing 12.5 µL of Type-it HRM PCR Master Mix (Qiagen, Hilden, Germany) with the EvaGreen fluorescent marker and 17.5 pmol of each pair of primers. The amplification temperature profile included an initial denaturation of 95 °C/5 min and 45 amplification cycles consisting of 95 °C/10 s denaturation, 55 °C/30 s annealing, and 72 °C/10 s elongation. Subsequent HRMA analysis was performed at 0.05 °C/s in the range of 65–95 °C and continuous fluorescence reading (20 reads/s). The evaluation of the reaction was made possible by comparing the melting curves with the standards for *Trichophyton rubrum* and *T. interdigitale*. Samples that were positive but did not show peaks consistent with the standards used were further identified by sequencing analysis [50,51].

#### 2.2.3. Onychomycosis Therapy

The cohort of 40 patients was divided into three groups receiving either (a) NPAR alone (*n* = 12), (b) simultaneous treatment with antimycotics (ATM) and NTP (*n* = 11), or (c) initial treatment with NPAR and then repeated exposure to NTP (*n* = 17). The protocols of all treatment and follow-up procedures are described schematically in the flowchart of the study design (see Figure 4).

#### 2.2.4. Nail Plate Abrasion and Refreshment

A group of patients was provided with NPAR. Sandpaper was attached to a dermabrader whose speed had to be appropriately controlled. The sandpaper was moved in vertical, horizontal, and diagonal directions. The sanding did not require anesthesia and was done on the nail edges. Care was taken to not cause any periungual pain or lesions, and no damage to the nail bed occurred. This procedure leads to refreshment of the nails because it aids in thinning the nail plate and, with subungual debridement of the hyperkeratosis, decreases the critical fungal mass (see Figure 5f). At the same time, the procedure also facilitates the action of the other therapeutic tool, namely NTP. The clinical condition of the nails before and after the procedure is shown in Figure 5a–d. A case of ingrown toenails with onychomycosis is shown in Figure 5e–g.

#### 2.2.5. Antimycotic Therapy

This treatment included conventional application of topical preparations ciclopirox olamine (ointment or nail polish), amorolfine, econazole, or urea-containing ointments (40%). In two cases, terbinafine tablets were administered intermittently for two cycles of 250 mg per day for two weeks followed by four weeks off treatment.

#### 2.2.6. Therapeutic NTP Exposure of Nails

The arrangement of the NTP-producing apparatus was identical to that used for NTP exposure of the plates; the actual therapeutic nail exposure arrangement is shown in Figure 6. Based on previous experiences [34,38,42] and the in vitro experiment in the first part of this work, patients underwent 16 exposures lasting 20 min on a schedule of two exposures per week during two months.

#### 2.2.7. Evaluation of Therapy

The therapy was completed and evaluated after six months. The clinical therapeutic effect was estimated subjectively. Restoration of more than 90% of the affected nail plate was classified as a clinical cure. No restoration was classified as no clinical improvement and restoration to 25–50% of the normal plate as moderate clinical improvement. The clinical cure was defined as a clinically normal nail with a mean increase in clear nail of 3–5 mm from baseline to the final follow-up at 6 months. To evaluate the clinical efficacy of NTP treatment, high-quality photographs were taken and evaluated in an independent dermatology center, with the evaluating dermatologist being blinded as to which patient was treated by which method. Objective evaluation was performed using the diagnostic methods mentioned above (microscopy, cultures of nail samples, etc.). Negative KOH microscopy and culture after treatment were considered as a mycological cure. All possibilities and combinations of efficacy measures for topical and oral onychomycosis treatment are summarized in Table 2. Examples of nail plates with 50% involvement are shown in Figure 7a, 75% involvement in Figure 7b, moderate clinical improvement in Figure 7c, and an almost complete cure in Figure 7d.

The statistical significance of completely cured and almost completely cured patients treated by NPAR with NTP was obtained by Fisher’s exact test.

## 3. Results

### 3.1. In Vitro Experiments

The time-dependent relationship between *Trichophyton interdigitale* inactivation and its growth status is summarized in Figure 8a–e. It is apparent that exposure to NTP at the time of inoculation and repeated in the following days completely inhibited fungal growth (Figure 8a). Although the growth of the control culture was not yet apparent, the exposure applied three days later and on subsequent days caused only partial inhibition (Figure 8b). The following figures (Figure 8c,d) show that the growth inhibition efficiency further decreased as the first day of exposure was postponed; the exposure on the sixth day after inoculation (Figure 8e) had a negligible inhibitory effect, if any. The experiment performed under the same conditions with *Trichophyton rubrum* and *Trichophyton benhamiae* gave almost identical results. Furthermore, the graphs analogous to the above *T. interdigitale* experiment are almost identical and are therefore not shown.

### 3.2. Onychomycosis Clinical Study

#### 3.2.1. Participant Characteristics

The cohort of 40 patients included in this study was gender balanced, with 47.5% males and 52.5% females. In all patients, more than 50% and 75% of their nail plates were affected by onychomycosis (see Figure 7a,b). In 22 patients, *T. rubrum* was identified by culture while the remaining five patients had cultures positive for *T. interdigitale*. Thirty cases were microscopically positive for dermatophytes in the nail samples, and ten patients displayed extensive onycholysis of their nail plates.

#### 3.2.2. Nail Plate Abrasion and Refreshment (Protocol 1)

Table 3 summarizes patients treated exclusively with NPAR according to Protocol 1. All patients were positive for the presence of dermatophytes; the fungal etiological agent did not disappear in any of them. Moderate clinical improvement was noted in 50% of the cases, and no effect was observed in the remaining ones.

#### 3.2.3. Combination of Antimycotics with Non-Thermal Plasma Treatment (Protocol 2)

Table 4 presents patients treated concurrently with conventional topical and systemic ATM and NTP according to Protocol 2. It shows similar findings to Table 3: *T. rubrum* was found in all but one case. DNA/PCR was not performed due to technical difficulties. Although the varying type of ATM therapy may seem as unsystematic and, in the comparison, it cannot be considered as significant, the overall therapeutic effect was even less satisfactory than in the previous case (36.4% moderate clinical improvement (MCI) and 9.1% mycological cure (MC)).

#### 3.2.4. Combination of Nail Plate Abrasion and Refreshment with Non-Thermal Plasma Treatment (Protocol 3)

The results presented in Table 5 are related to a therapeutic procedure combining NPAR and NTP according to Protocol 3. Although in some cases, the etiological agent was detected only by microscopy and not by culture, it was eliminated in six out of seven positive cases. The PCR-positive cases after therapy may be explained by the presence of inactivated fungal residues. A complete cure was achieved in 28.5%, almost complete cure in 28.5% and mycological cure in 28.5% of seven confirmed onychomycosis cases. One case was evaluated as no clinical improvement. In Patient 26, a relapse of mycological positivity (confirmed by KOH microscopy and culture) occurred 30 days after almost successful therapy; other mycologically negative patients did not relapse.

The results of patients with dermatoscope confirmed/clinically positive findings of onycholysis but negative mycological examination and treated with the same therapeutic procedure (NPAR and NTP according to Protocol 3) are shown separately in Table 6. Clinical cure was achieved in 30%, moderate clinical improvement in 30%, and no improvement in 40% of 10 cases of clinical onychomycosis.

#### 3.2.5. Evaluation of the Results

The above results are summarized in a simple graph in Figure 9. The treatment using NPAR alone resulted in moderate clinical improvement in 50% of patients. The topical antimycotic treatment was associated with moderate clinical improvement in 36.4% of cases. Topical therapy with NPAR and NTP at the same time led to some improvements characterized as moderate clinical improvement, clinical cure, mycological cure, almost complete cure, and complete cure, with a total benefit of 70.6% in the treatment of onychomycosis.

Mycological cure effects of NPAR alone, topical ATM combined with NTP, and NPAR combined with NTP are visualized separately in Figure 10. After six months, the mycological cure effects of NPAR alone and topical ATM combined with NTP were negative. In the case of NPAR with NTP, mycological cure was achieved in 85.7% of patients.

The numbers of completely cured and almost completely cured patients treated by NPAR with NTP are statistically higher than the numbers of those treated by NPAR or ATM with NTP, at a significance level close to 5%. Moreover, the number of completely cured patients treated by NPAR with NTP is statistically higher than the numbers of those treated by the other two methods together at a significance level close to 1%. These results were obtained by Fisher’s exact test with the R software (*p*-value = 0.05211). This test originally designed for 2 × 2 contingency tables can be applied for our data, e.g., for testing that the true proportion of completely cured and almost completely cured patients treated by NPAR with NTP is greater than the true proportion of those treated by NPAR in the case of a small sample size. The computed *p*-values are slightly higher than 5 %, but Fisher’s exact test is rather conservative, so that the real significant level could be smaller than these *p*-values.

## 4. Discussion

The sensitivity to NTP was previously found to be considerably variable among different fungal species: while *Trichophyton* spp. appeared to be highly sensitive, exposure of *Aspergillus* and *Penicillium* spp. to NTP had little or no effect. An earlier study [33] showed no inhibition of these species, only a certain slowdown of their growth. A dramatically different effect of NTP on fungal inactivation was also observed in a study already mentioned in the Introduction [31]. This variability is probably mainly due to the different experimental arrangement of plasma sources and their different efficacy. The variable sensitivity of the exposed strains may also play an important role, as may the different nature of the substrates on which the reported values were measured. In general, good efficacy of NTP was observed in the genus *Trichophyton*, suggesting a good basis for effective treatment of mycoses caused by dermatophytes [54]. Even in these cases, however, the results presented in this study showed that NTP is mainly effective when used in the early stages of growth while exposure of later and sporulated forms is less effective. This corresponds to another pilot study on NTP in 19 patients with toenail onychomycosis showing a clinical cure rate of 53.8% and a mycological cure rate of 15.4% [55].

These findings may also explain the mechanism of the therapeutic effect of NTP. Its application effectively prevents the development of “young” molds, while developed molds are scarcely affected. In the latter cases, hyperkeratosis needs to be treated first by other means such as NPAR. This supports the use of Protocol 3 of this study associated with mycological cure in 85.7% of seven patients with toenail onychomycosis, as compared with two other therapeutic approaches (Protocols 1 and 2). Thus, the long-term application of NTP considerably prevents the development or re-development of early forms of the etiological agent and recurrence of the disease. Nail abrasion is considered to be a practical method for stimulating nail refreshment, helpful in removing the essential part of the affected nail and not too invasive to the nail bed [56]. In combination with subungual debridement of hyperkeratosis and aeration of the whole nail body at the same time, it allows access to those areas of the nail plate where other follow-up therapeutic methods could work better.

As for experimental therapeutic application in human onychomycosis, the exposure to NTP appeared useful in combination with NPAR. Unfortunately, even with this arrangement, complete or at least partial success was achieved in only some cases. Nevertheless, it should be noted that the traditional methods were not successful in treating any of the cases and caused only moderate improvement in less than half of them. The only explanation of this phenomenon is insufficient penetration of the active substances through the nail tissue barrier and into dermatophyte particles.

NTP was previously confirmed as a safe therapeutic method for use in dermatology [55,57]. The combination of NPAR and NTP treatment was also relatively painless and comfortable, with all patients reporting the sessions as tolerable. This approach prevents drug-drug interactions and serious adverse events connected with systemic treatments. The treatment courses are also considerably shorter, ensuring better patient compliance as compared with topical treatments. Many patients are unable to properly care for their nails, for example due to other health conditions.

This contribution shows that NTP technologies are not only applicable in skin dermatomycosis therapy [37,38,58], but could also be useful in the treatment of onychomycosis. To further improve the outcomes, it may be useful to use alternative plasma sources or to increase the efficacy of the currently used source. The results should be verified in a larger population of patients with a wider age range. The limitations of this study include poor knowledge of the penetration of active NTP particles into various nail and subungual structures containing dermatophytes. Solving this problem may enable better targeting of onychomycosis therapy. The limit of the study can also be considered to be the shortened time for evaluating the effects of therapies, which should be performed after 18 months. On the other hand, ninety percent restoration of nails at 3–5 mm of growth in six months indicates a promising development towards the elimination of the onychomycosis agent. This is also confirmed by the results according to the mycological cure effect.

## 5. Conclusions

The dynamics of fungal inactivation with non-thermal plasma was determined. The dermatophytes *Trichophyton* spp. are mainly inactivated in the early phases of their growth. Even sensitive dermatophytes must be exposed to plasma in the early growth phase; exposure after six days of growth is no longer effective. Comparison of various methods for treating human onychomycosis caused by dermatophytes showed that the combination of nail plate abrasion and refreshment with non-thermal plasma was beneficial in more than 70% of patients. This study also showed that the synergistic effect of NPAR and NTP has an unquestionable effect on mycological cure of the affected nail plate.

## Figures and Tables

**Figure 1 jof-06-00214-f001:**
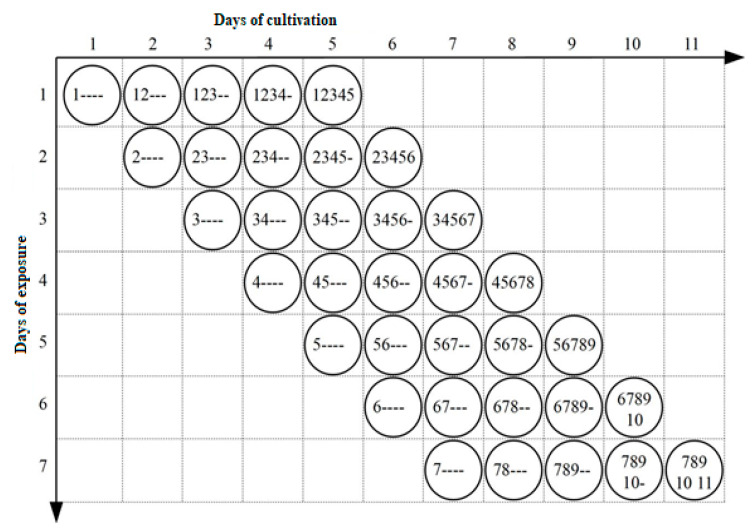
Scheme of time-dependent exposure to non-thermal plasma. Numbers denote the day of exposure; symbol ‘-’ indicates no further exposure.

**Figure 2 jof-06-00214-f002:**
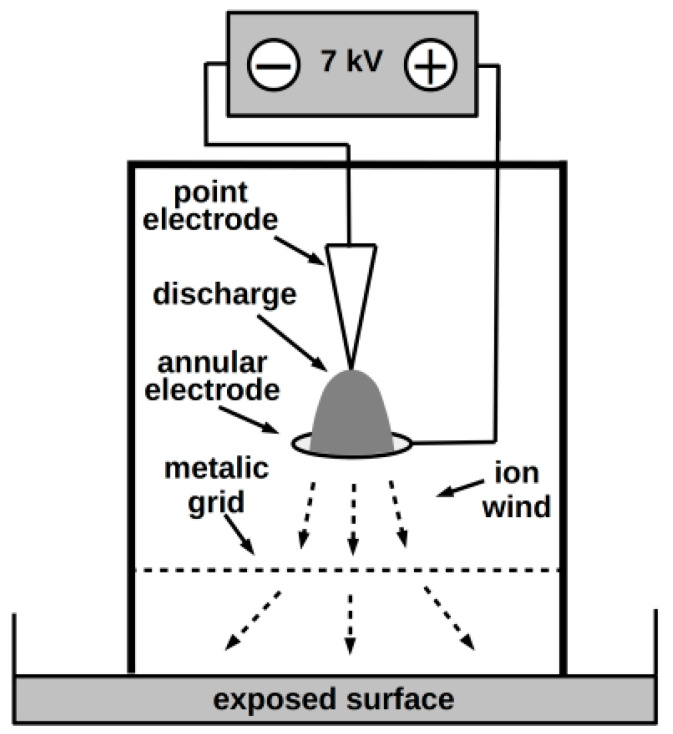
Scheme of the NTP source.

**Figure 3 jof-06-00214-f003:**
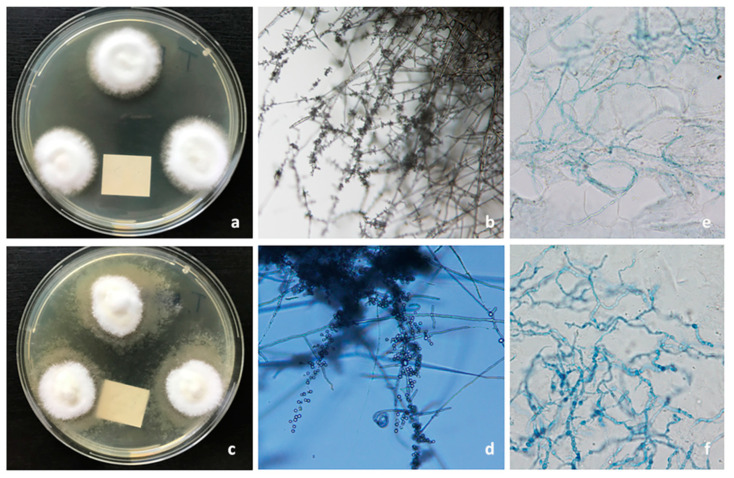
Mycological examination: (**a**) *T. rubrum* colonies, 10-day-old culture on malt extract agar incubated at 26 °C. (**b**) *T. rubrum* macro- and micro-conidia in microculture on Sabouraud agar (200× magnification). (**c**) *T. interdigitale* colonies, 10-day-old culture on malt extract agar incubated at 26 °C. (**d**) *T. interdigitale* mycelium and microconidia in microculture on Sabouraud agar (200× magnification). (**e**) Microscopic image of *T. rubrum* dermatophytic hyphae in the nail samples (KOH+Myko-Ink stain, 400× magnification). (**f**) Microscopic image of *T. interdigitale* dermatophytic hyphae in the nail samples (KOH+Myko-Ink stain, 400× magnification).

**Figure 4 jof-06-00214-f004:**
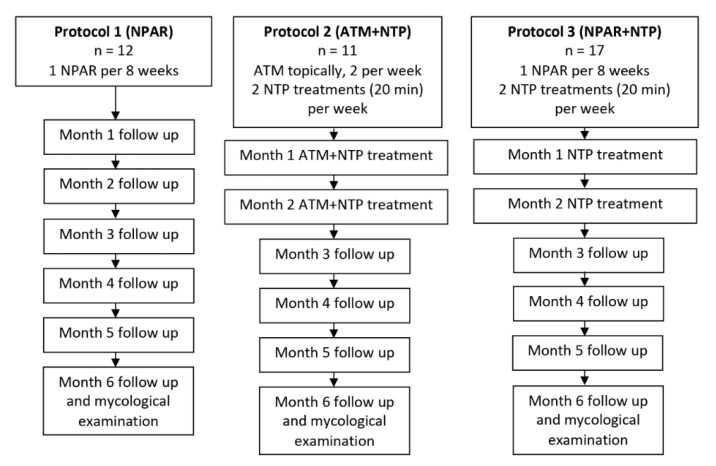
Flowchart of the study design. NPAR, nail plate abrasion and refreshment; ATM, antimycotics.

**Figure 5 jof-06-00214-f005:**
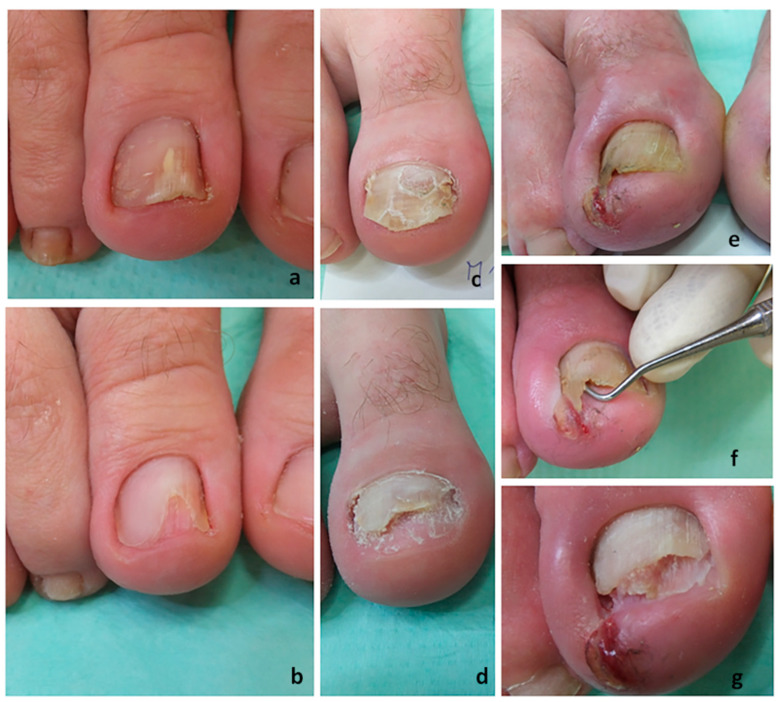
Nail plate abrasion and refreshment: (**a**) A big toenail affected by 50% involvement before NPAR. (**b**) The big toenail at the end of the NPAR procedure. (**c**) A big toenail affected by 75% involvement before NPAR. (**d**) The big toenail at the end of the NPAR procedure. (**e**) An ingrown big toenail affected by onychomycosis. (**f**) Refreshment and collecting subungual debris after abrasion. (**g**) The big toenail at the end of the NPAR procedure.

**Figure 6 jof-06-00214-f006:**
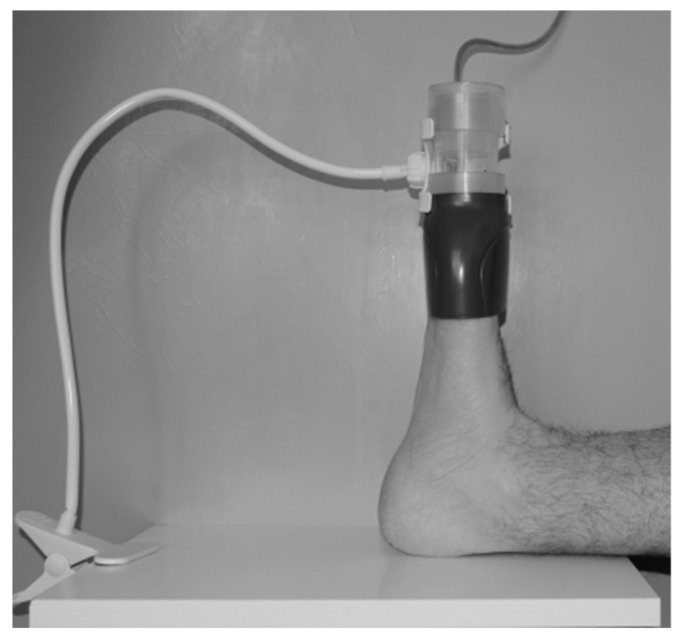
Exposure of affected toenails.

**Figure 7 jof-06-00214-f007:**
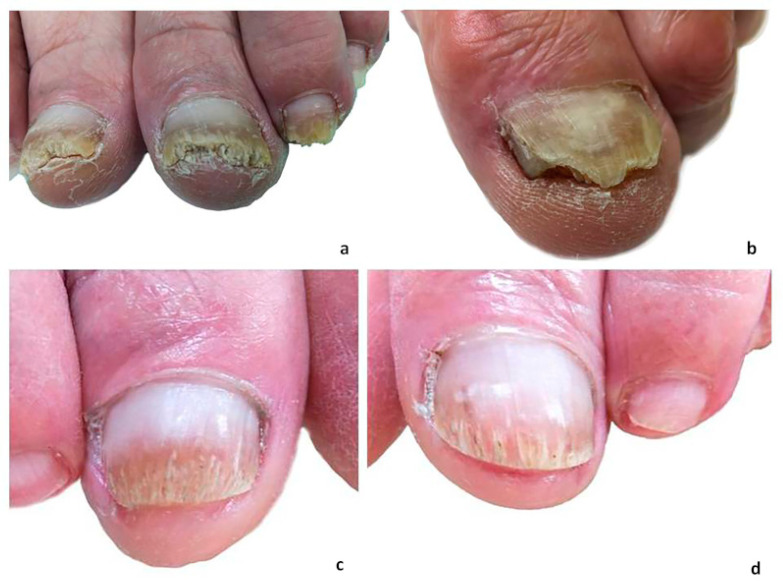
Examples of the therapeutic effect of NTP in combination with nail hygiene in vivo: (**a**) Big toenails of both feet with onychomycosis (*T. rubrum*) before NTP application, hyperkeratosis of the distal parts of the nail plates with crumbling, growth of the healthy parts proximally about 3–4 mm, other toenails also distally affected by onychomycosis to varying extents. (**b**) A big toenail with onychomycosis (*T. interdigitale*) before NTP application; the uneven distal part of the nail plate shows signs of hyperkeratosis and fungal appearance; there are signs of cleft on the medial side of the nail plate. (**c**,**d**) The big toenails of both feet at six months from the first NTP application with no hyperkeratosis of the distal parts; microscopy and culture of the nail samples were negative.

**Figure 8 jof-06-00214-f008:**
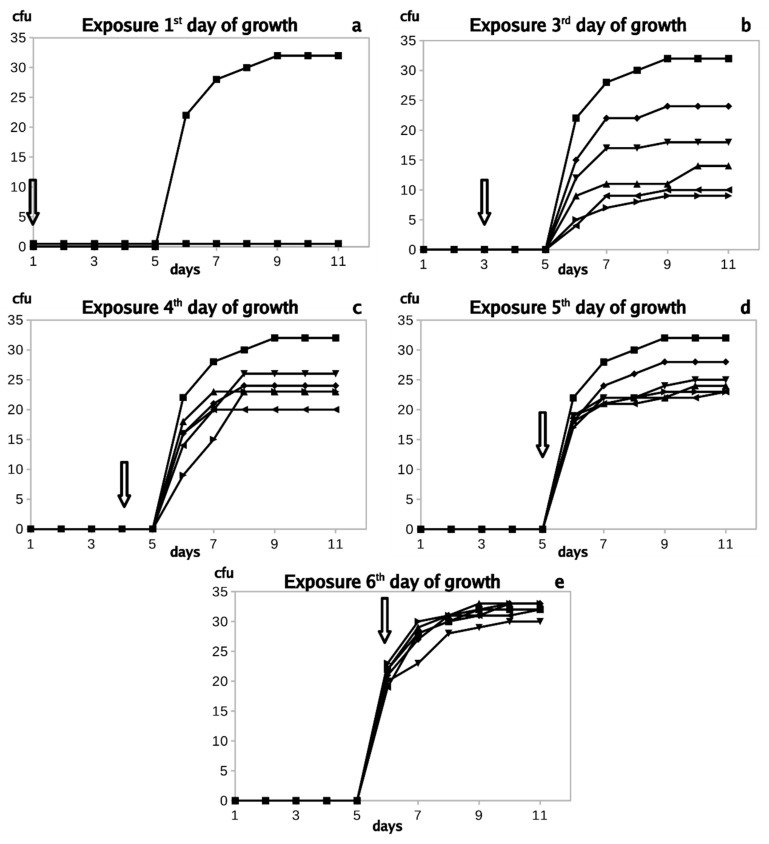
Colony-forming units (cfu) of *Trichophyton interdigitale* exposed on the first (**a**), third (**b**), fourth (**c**), fifth (**d**), and sixth (**e**) day after inoculation. The lines are interlaced as a moving average. The upper line and full rings represent unexposed culture; the lower curves (coinciding with the x-axis) represent experimental series. (**a**) 1----, 12---, 123--, 1234-, and 12345; (**b**) 3----, 34---, 345--, 3456-, and 34567; (**c**) 4----, 45---, 456--, 4567-, and 45678; (**d**) 5----, 56---, 567--, 5678-, and 56789; (**e**) 6----, 67---, 678--, 6789-, and 678910. Error bars are not inserted for better picture clarity. An arrow marks the NTP exposure.

**Figure 9 jof-06-00214-f009:**
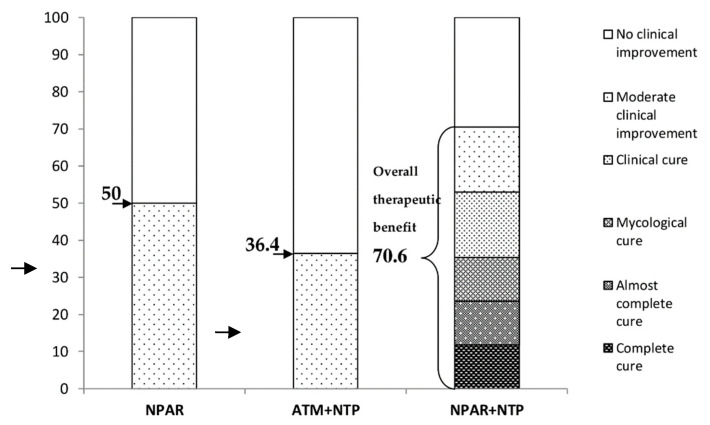
Therapeutic effect (%) in particular groups of patients.

**Figure 10 jof-06-00214-f010:**
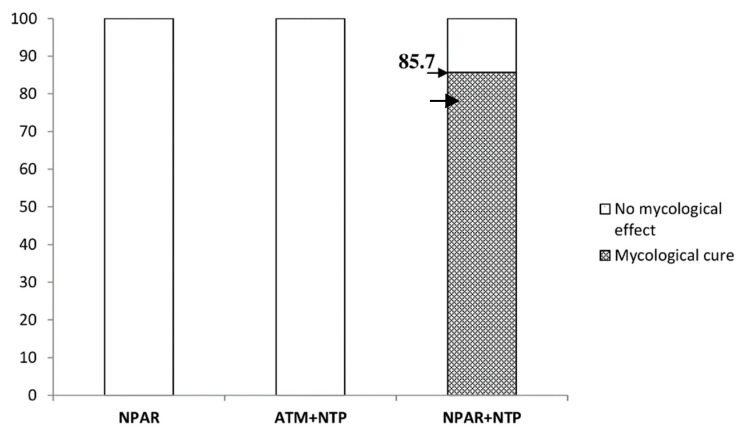
Mycological cure effect (%) in particular groups of patients.

**Table 1 jof-06-00214-t001:** Sequences of primers and probes used for real-time PCR.

Primer/Probe	Region	Sequence 5′-3′	Reference
PanDerm_F1	ITS1-5,8S	AGCGCYCGCCGRAGGA	[52]
PanDerm_R1	ITS1-5,8S	GATTCACGGAATTCTGCAATTCAC	[52]
Derm_FAM	ITS1-5,8S	[6FAM]CGCATTTCGCTGCGTTCTTCATC[BHQ1]	[52]
PanDerm_HRM_F	ITS1-5,8S-ITS2	TGCGGAAGGATCATTAACG	[50]
PanDerm_HRM_R	ITS1-5,8S-ITS2	ACCAAGAGATCCGTTGTTG	[50]

6FAM: fluorescein; BHQ1: black hole dark quencher.

**Table 2 jof-06-00214-t002:** Efficacy measures for topical and oral treatment of onychomycosis. Clinical and mycological endpoint evaluation criteria were taken from a study by Gupta and Studholme [53] and modified for the purposes of this study.

Endpoint (Mycological/Clinical)	Definition
Mycological cure	Negative KOH microscopy, negative culture
Clinical cure	>90% clearance of the previously affected part of the nail plate
Moderate clinical improvement	25–50% (or 30–60%) clinical improvement
No clinical improvement	0–25% (or 0–30%) clinical improvement
**Endpoint (Mycological + Clinical)**	
Complete cure	Mycological cure, clinical cure (0% nail plate involvement)
Almost complete cure	Mycological cure, ≤10% nail plate involvement (or ≤5% nail plate involvement

**Table 3 jof-06-00214-t003:** Therapeutic effect of nail plate abrasion and refreshment, Protocol 1.

	Examination before Therapy			Examination after Therapy			
Patient No.	Microscopy	Culture	DNA/PCR	Microscopy	Culture	DNA/PCR	Therapeutic Effect
1	positive	TI	TI	positive	TI	TI	MCI
2	positive	TR	TR	positive	TR	TR	MCI
3	positive	TI	TI	positive	TI	TI	MCI
4	positive	TR	TR	positive	TR	TR	MCI
5	positive	TR	TR	positive	TR	TR	MCI
6	positive	TR	TR	positive	TR	TR	MCI
7	positive	TR	TR	positive	TR	TR	NCI
8	positive	TR	TR	positive	TR	TR	NCI
9	positive	TR	TR	positive	TR	TR	NCI
10	positive	TR	TR	positive	TR	TR	NCI
11	positive	TI	TI	positive	TI	TI	NCI
12	positive	TR	TR	positive	TR	TR	NCI

TR: *Trichophyton rubrum*; TI: *Trichophyton interdigitale*; MCI: moderate clinical improvement; NCI: no clinical improvement.

**Table 4 jof-06-00214-t004:** Therapeutic effect of ATM combined with NTP, Protocol 2.

	Examination before Therapy			Examination after Therapy			
Patient No.	Microscopy	Culture	DNA/PCR	Microscopy	Culture	Therapeutic Effect	ATM
13	positive	TR	NA	positive	TR	MCI	Amo
14	positive	TR	NA	positive	TR	MCI	Amo, Eco, Cic
15	positive	TR	NA	positive	TR	MCI	Amo
16	positive	TR	NA	positive	TR	MCI	Cic, 40% urea/vas
17	positive	TR	NA	positive	TR	NCI	Cic NP, Ter tbl
18	positive	TR	NA	positive	TR	NCI	Amo NP, Cic
19	positive	TR	NA	positive	TR	NCI	Amo NP, Cic
20	positive	TR	NA	positive	TR	NCI	Amo NP
21	positive	TR	NA	positive	TR	NCI	Cic, Amo NP
22	positive	TR	NA	positive	TR	NCI	Ter tbl, Cic NP
23	positive	negative	NA	positive	negative	NCI	Cic

Amo: amorolfine; Amo NP: amorolfine nail polish; Eco: econazole; Cic: ciclopirox olamine; Cic NP: ciclopirox olamine nail polish; Ter tbl: terbinafine tablets (2 cycles of 250 mg per day for 2 weeks followed by 4 weeks off treatment); urea/vas: urea containing ointments; NA: not applied; TR: *Trichophyton rubrum*; MCI: moderate clinical improvement; NCI: no clinical improvement.

**Table 5 jof-06-00214-t005:** Therapeutic effect of NPAR combined with NTP, Protocol 3.

	Examination before Therapy			Examination after Therapy			
Patient No.	Microscopy	Culture	DNA/PCR	Microscopy	Culture	DNA/PCR	Therapeutic Effect
24	positive	negative	negative	negative	negative	negative	Complete C
25	positive	TR	TR	negative	negative	negative	Complete C
26	positive	TI	TI	negative	negative	TI	ACC
27	positive	negative	negative	negative	negative	negative	ACC
28	positive	TI	TI	negative	negative	negative	MC, NCI
29	positive	TR	TR	negative	negative	TR	MC, NCI
30	positive	TR	TR	positive	TR	TR	NCI

TR: *Trichophyton rubrum*; TI: *Trichophyton interdigitale*; Complete C: complete cure; ACC: almost complete cure; MC: mycological cure; NCI: no clinical improvement.

**Table 6 jof-06-00214-t006:** Therapeutic effect of NPAR combined with NTP, Protocol 3, in patients with clinical signs of onycholysis but negative mycological examination.

	Examination before Therapy			Examination after Therapy			
Patient No.	Microscopy	Culture	DNA/PCR	Microscopy	Culture	DNA/PCR	Therapeutic Effect
31	negative	negative	negative	negative	negative	negative	Clinical C
32	negative	negative	negative	negative	negative	negative	Clinical C
33	negative	negative	negative	negative	negative	negative	Clinical C
34	negative	negative	negative	negative	negative	negative	MCI
35	negative	negative	negative	negative	negative	negative	MCI
36	negative	negative	negative	negative	negative	negative	MCI
37	negative	negative	negative	negative	negative	negative	NCI
38	negative	negative	negative	negative	negative	negative	NCI
39	negative	negative	negative	negative	negative	negative	NCI
40	negative	negative	negative	negative	negative	negative	NCI

Clinical C: clinical cure; MCI: moderate clinical improvement; NCI: no clinical improvement.

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
