# Peer review of "Inactivation of Dermatophytes Causing Onychomycosis and Its Therapy Using Non-Thermal Plasma"

_jof, 2020, doi:10.3390/jof6040214_

Round 1

Reviewer 1 Report

line 67 I would say nail debridement instead of "pedicure procedure".  Pedicures are more cosmetic and nail debridement describes reduction of length and thickness of affected nail

line 200: I understand how you chose via dermatoscope, but it is standard to say whether the nails were mild-moderate-severe. ie the onychomcyosis severity index.  you don't mention the involvement until line 351--so I suggest you label these nails as moderate to severe earlier in the paper.

line 289  it would have been a better protocol to choose only one type of antimycotic therapy.  these have varying levels of efficacy and transmission (oral vs topical).  This is a large flaw as you can't compare this--it's like comparing apples to oranges.  The two things that should be compared in this study is NPAR alone to NPAR plus plasma.  

To me, Table 4 is a mess and doesn't add value since there were so many therapeutics used--and more than one in some patients.  This was not well done in a very well structured paper.  

Table 5--why include patients who had positive KOH, but negative culture and PCR?  the KOH is just showing the presence of hyphae, but not viable substance.  the culture and PCR would confirm that the KOH was a true positive. 

Author Response

Dear dear reviewer

Thank you for your insightful comments and recommendations! Enclosed, please find our revise manuscript with responses (R:) to your comments ordered sequentially. Changes are highlighted in the revised manuscript.

line 67 I would say nail debridement instead of "pedicure procedure".  Pedicures are more cosmetic and nail debridement describes reduction of length and thickness of affected nail

R: The text has been adapted in line with the recommendation.

line 200: I understand how you chose via dermatoscope, but it is standard to say whether the nails were mild-moderate-severe. ie the onychomcyosis severity index.  you don't mention the involvement until line 351--so I suggest you label these nails as moderate to severe earlier in the paper.

R: The text has been adapted in line according to the recommendation.

line 289  it would have been a better protocol to choose only one type of antimycotic therapy.  these have varying levels of efficacy and transmission (oral vs topical).  This is a large flaw as you can't compare this--it's like comparing apples to oranges.  The two things that should be compared in this study is NPAR alone to NPAR plus plasma.

R: see bellow

To me, Table 4 is a mess and doesn't add value since there were so many therapeutics used--and more than one in some patients.  This was not well done in a very well structured paper.  

R: We agree that it should be better to choose only one type of antimycotic therapy. However, for this study we didn‘t have so many suitable patients. We suggest to include this one group at least to demonstrate that the NPAR plus plasma is more effective than antimycotic therapy although it cannot be conseder as significat. We have add explanation text to the results to section 3.2.3. Combination of antimycotics with non-thermal plasma treatment.

Table 5--why include patients who had positive KOH, but negative culture and PCR?  the KOH is just showing the presence of hyphae, but not viable substance.  the culture and PCR would confirm that the KOH was a true positive. 

R: We agree, however the positivity of direct microscopy of nail samples in onychomycosis indicates the presence of a dermatophyte and in general it cannot be consider as inactive. Usually the cultivations of so heterogeneous nail samples are positive in about 70 % cases only, even in the case of progressive nail plate involvement. In our clinical practice, the microscopy has higher yield than the cultivation. Dermatophytes are less culturable in vitro.

Reviewer 2 Report

The authors performed a study of nonthermal plasma application on onychomycosis and used a culture model to find out some basics of NTP on T rubrum and T interdigitale. They found that these dermatophytes when seed onto plates have to be treated with NTP within the first 3 days, later the growth in culture was only minimally or not inhibited. Preparing the nails with dermabrasion improved the results whereas NTP alone did almost not work at all.

From several phrases in the text there is the impression that the authors believe that the nail plate is the main site of the fungi in onychomycosis. This is a fundamental misunderstanding of most onychomycoses except for the rare superficial type.

Page 2, line 53: anthropophilic mycoses are asymptomatic, zoophilic often intensely inflammatory.

Page 2, lines 72-74: Lasers have not been proven to be efficacious by experienced mycology groups. 40 -50% heating with the laser: Who can tolerate more than 45° on the nailbed or matrix?  More than an hour of 55° is needed to kill dermatophytes.

Page 2, lines 88-94, page 3 lines 107-108: You cannot refer to a number of publications to explain how something is working!

Page 3, line 146: 16 exposures of 20 min each?

Page 8, Fig 4: What is NH?

Page 8, line 277: Dermabrasion decreases the fungal load? This would mean that the relevant fungal mass is relatively superficially in the nail plate, which is absolute not true.

Page 8, line 291: Why is giving terbinafine in onychomycosis off-label? Or is it not approved in Czechia?

Page 9, last line: 16 exposures every 3 days to 1 week means almost up to 4 months of treatment!

Page 10, line 300 ff: A final evaluation after 6 months is too early. The big toenail requires 18 months to grow out fully, so after 6 months only one third may have grown out. How can this be evaluated as 90% restoration. On the other hand, the authors write that 3 – 5 mm normal nail from normal

Page 11, lines 330 ff: It is evident that NTP works best when applied early, after 6 days in culture there was only a negligible effect. However, in onychomycosis, the fungi have been present for months if not years, they are in their optimal habitat, and they are not on or in but under the nail plate.

Page 17, line 456: Is it possible to increase the efficacy of NTP with your device? You already use 7000 V and 0.15 A, which gives a considerable wattage.

Page 17, lines 458 ff: “The limitations of this study include poor knowledge of the penetration of active NTP particles into various nail structures containing dermatophytes. Solving this problem may enable better targeting of onychomycosis therapy.” Actually, almost all pathogenic fungi are under the nail and the overlying nail plate is rather a barrier than the preferred nutrient for dermatophytes.

Author Response

Dear dear reviewer

Thank you for your insightful comments and recommendations! Enclosed, please find our revise manuscript with responses (R:) to your comments ordered sequentially. Changes are highlighted in the revised manuscript.

General: From several phrases in the text there is the impression that the authors believe that the nail plate is the main site of the fungi in onychomycosis. This is a fundamental misunderstanding of most onychomycoses except for the rare superficial type.

R: We agree, therefore we have specified the NPAR methodology in several passages of the text. It includes also subungual debridment of hyperkeratoses and nail body aeration to achieve a better effect of NTP.

Page 2, line 53: anthropophilic mycoses are asymptomatic, zoophilic often intensely inflammatory.

R: The meaning of the sentence has been modified: In the text it means infections in animals, which can occur directly on animals and take place asymptomatically.

Page 2, lines 72-74: Lasers have not been proven to be efficacious by experienced mycology groups. 40 -50% heating with the laser: Who can tolerate more than 45° on the nailbed or matrix?  More than an hour of 55° is needed to kill dermatophytes.

R: We agree and modify the text, this therapy is presented as unsuitable for patients therapy.

Page 2, lines 88-94, page 3 lines 107-108: You cannot refer to a number of publications to explain how something is working!

R: We have describe the plasma, its generation and possible applications in more details. However, this is very range area and the simplification is very difficult. The wide but still brief review should take one full paper.

Page 3, line 146: 16 exposures of 20 min each?

R: In the text, the sentence has been simplified.

Page 8, Fig 4: What is NH?

R: The incorrect inscription in the diagram was removed from Figure 4.

Page 8, line 277: Dermabrasion decreases the fungal load? This would mean that the relevant fungal mass is relatively superficially in the nail plate, which is absolute not true.

R: We agree, it really looks like that. In several passages of the text, we have explain it and specified the NPAR methodology, which also includes subungual debridment of hyperkeratoses and nail body aeration to achieve a better effect of NTP.

Page 8, line 291: Why is giving terbinafine in onychomycosis off-label? Or is it not approved in Czechia?

R: Terbinafine was administered in pulses due to fear of contraindications with other drugs used by patients. This procedure is not standard, but it is true that it is redundant information and has been removed from the text. Of course, the drug is registered in the Czech Republic.

Page 9, last line: 16 exposures every 3 days to 1 week means almost up to 4 months of treatment!

R: There was an error by editing the text, thank you very much for your warning. We adjusted according to the actual exposure, as in the diagram in Figure 4.

Page 10, line 300 ff: A final evaluation after 6 months is too early. The big toenail requires 18 months to grow out fully, so after 6 months only one third may have grown out. How can this be evaluated as 90 % restoration. On the other hand, the authors write that 3 – 5 mm normal nail from normal

R: Thank you very much for this comment, it is crucial. In the evaluation, it was thought of as 90% of restoration from a given nail plate disability, the discs were affected moderately. However, in the limits of the study a shortened time for evaluating the results of the clinical effect of the therapy was mentioned in the discussion, an 18-month observation is actually more appropriate.

Page 11, lines 330 ff: It is evident that NTP works best when applied early, after 6 days in culture there was only a negligible effect. However, in onychomycosis, the fungi have been present for months if not years, they are in their optimal habitat, and they are not on or in but under the nail plate.

R: And for this reason, we perform NPARs and try to thoroughly clean the nail from the affected areas, also under the nail plate, leaving only the remnants of dermatophytes and allow free way for NTP particles, which obviously works. However, it is true that this may fail and relapse. It may occur also from the interdigital sites, however, in these areas, NTP should be even more effective, according to our experiments on animal and human skin lesions caused by dermatophytes.

Page 17, line 456: Is it possible to increase the efficacy of NTP with your device? You already use 7000 V and 0.15 A, which gives a considerable wattage.

R: The current is not 0.15 A but 0.15 mA so the wattage is in order of 1 mW. The increase of wattage is possible and may lead to decrease of exposure time or more effective treatment. However, this may be the aim for future study. However, this question is not so straightforward and depends on many parameters.

Page 17, lines 458 ff: “The limitations of this study include poor knowledge of the penetration of active NTP particles into various nail structures containing dermatophytes. Solving this problem may enable better targeting of onychomycosis therapy.” Actually, almost all pathogenic fungi are under the nail and the overlying nail plate is rather a barrier than the preferred nutrient for dermatophytes.

R: The text has been modified to point out that it is clear that the onychomycosis agent is mainly subungual.

Reviewer 3 Report

This is a very interesting work about the use of non-thermal plasma in the treatment of onychomycosis. The manuscript is written correctly and the work well-developed. In my opinion it can be accepted for publication after a minor review.

The authors must specify whether the clinical study was carried out on the toenails or, on the contrary, on the feet and hands.

Authors must specify in the manuscript the statistical treatment used to compare the samples.

Author Response

Dear dear reviewer

Thank you for your insightful comments and recommendations! Enclosed, please find our revise manuscript with responses (R:) to your comments ordered sequentially. Changes are highlighted in the revised manuscript.

The authors must specify whether the clinical study was carried out on the toenails or, on the contrary, on the feet and hands.

R: Thank you, it was specified at the beginning in the text of section 2.2.

Authors must specify in the manuscript the statistical treatment used to compare the samples.

R: The statistics was described in more details in the text.